# Effect and Mechanism of Solidified Microstructure on Deformation Behavior, Mechanical Properties, and Residual Stress of Cu-Ni-Si Alloy

**DOI:** 10.3390/ma15248724

**Published:** 2022-12-07

**Authors:** Wanneng Liao, Chenxing Zhang, Hui Qiang, Weifei Song, Hongwen Ren

**Affiliations:** 1College of Mechanical and Electrical Engineering, Nanjing University of Aeronautics and Astronautics, Nanjing 210016, China; 2College of Material Science and Technology, Nanjing University of Aeronautics and Astronautics, Nanjing 210016, China

**Keywords:** C70250 copper alloy, microstructure, deformation behavior, mechanical properties, residual stress

## Abstract

Cu-Ni-Si alloy is the key raw material for the lead frame of large integrated circuits. The disordered grain orientation of alloy billet, high hardening rate, residual stress, and poor surface quality of cold working strips seriously affect its processability. In order to improve the cold-working properties of Cu-Ni-Si alloy, two kinds of C70250 copper alloy strips were produced through hot mold continuous casting (HMCC) and cold mold continuous casting (CMCC) technology. The effects of solidified microstructure on the cold-working deformation behavior, mechanical properties, and residual stress of the alloy were studied. The results show that C70250 copper alloys with columnar grain and equiaxed grain were prepared through HMCC and CMCC. After a 98% reduction in cold rolling, columnar grain strip surface quality was very good, and the elongation was still as high as 3.2%, which is 2.9 times that of equiaxed grain alloy. The residual stress of equiaxed grain strips reached 363 MPa, which is 2.7 times that of columnar grain strips. During the cold rolling process, equiaxed grain strips are prone to cause intersecting plane dislocations, stacking faults, shear bands, and grain breakage during large deformation cold rolling. The columnar grain strip causes parallel plane dislocations, stacking faults, and shearbands. Furthermore, the deformation structure was found to be uniform, and, ultimately, the alloy formed a fibrous structure. Therefore, the elongation and latter distortion of columnar grain strips improved after being put through large deformation cold rolling, which greatly reduced residual stress.

## 1. Introduction

With the continuous development of electronic information, products are continuously becoming smaller, thinner, and more lightweight, demanding that frame materials have high strength, elasticity, and electrical conductivity, but also great capability for plasticity and formability. The mechanical properties, electrical conductivity, and residual stress distribution of Cu-Ni-Si alloy plates and strips are closely related to their microstructure [1,2]. In recent years, researchers in China and abroad have undertaken considerable efforts in the microalloying and process optimization of Cu-Ni-Si alloy, and its strength and electrical conductivity have been greatly improved. For example, Lei et al. [3] found that the addition of Al can refine the grain structure of Cu-Ni-Si alloy, promote the uniform dispersion distribution of precipitates, and improve the peak strength and stress relaxation resistance of the alloy. Gholami et al. [4] used large plastic deformation to significantly refine the microstructure of Cu-Ni-Si alloy, promote the appearance of precipitates, and delay the rearrangement of dislocations. The tensile strength and conductivity of the alloy are increased by 24% and 4%, respectively, when compared to Cu-Ni-Si alloy without plastic deformation. Ryu et al. [5] used a two-step rolling and aging process to increase the tensile strength of the alloy to 640 MPa and reduced its electrical resistivity to 1.475 × 10^−8^ Ωm by reducing the size of Ni_2_Si precipitates to 4 nm. Wu et al. [6] studied the complete aging process and the origin of property variation of Cu-2.69Ni-1.14Si-0.45Cr alloy. It was found that the pre-deformation not only speeds the aging process and promotes precipitation but also improves strength without any sacrifice of the plasticity or conductivity of the Cu-Ni-Si-Cr alloy.

Most of the above studies focus on the treatment effect of equiaxed grain Cu-Ni-Si alloy after undergoing aging treatment. However, cold working has a great influence on the properties of Cu-Ni-Si alloys with different microstructures. Cu-Ni-Si easily produces deformation structures, such as dislocation, deformation twins, and shearing of bands during large plastic cold-working deformation, resulting in large residual stress in cold-rolled sheets. This residual stress has a great impact on the properties and service life of the material. Eliminating or controlling residual stress in workpieces is an important scientific and engineering problem [7,8,9]. In randomly oriented equiaxed grain structures, the probability and number of deformation twins and shear bands is different in each grain during deformation, resulting in serious uneven deformation between grains [10,11]. These internal uneven deformations easily lead to high work hardening rates, residual stress, significant decreases in plasticity, and high amounts of stress accumulated at the edge of the plate during rolling. If the stress is not effectively reduced or removed, it is easy for stress accumulation to occur at the edge of the plate in the subsequent rolling process. When the breaking strength of the material is exceeded, this will lead to fracture. The equiaxed grain structure of Cu-Ni-Si alloy is refined to the submicron level after large plastic cold-working deformation [12]. The strength of the alloy is increased by grain boundary strengthening (Hall–Petch effect) and dislocation strengthening, but the strain caused by lattice defects is significantly enhanced, resulting in higher residual stress [13,14]. The columnar grain structure possesses high orientation, and each columnar grain will undergo elastic and plastic deformation together, which greatly reduces the stress between grains caused by strain disharmony and is conducive to improving elongation at room temperature [10,15,16,17]. Additionally, the C70250 alloy is an age-strengthened Cu-Ni-Si alloy with high strength, electrical conductivity, and stress relaxation resistance. It is very suitable as a lead frame alloy and for high-density integrated circuit packaging.

In the present work, C70250 copper alloy strips with equiaxed and columnar grain structures are prepared via cold mold continuous casting (CMCC) and hot mold continuous casting (HMCC) [18], respectively. Alloy strips with different microstructures are rolled with different deformation amounts, and the effects of different solidification structures on the deformation behavior, mechanical properties, and residual stress of C70250 copper alloy strips are studied. The relevant mechanism is revealed, which provides a theoretical basis for improving the cold-working properties of Cu-Ni-Si alloy strips.

## 2. Experimental Materials and Methods

### 2.1. Preparation and Cold Rolling of Strip

The Cu-2.79Ni-0.58Si-0.1Mg wt.% alloy (C70250 copper alloy) was vacuum-smelted in a medium-frequency induction furnace with argon protection. The liquidus and solidus of the sample were 1089 °C and 1054 °C, respectively, as observed via differential scanning calorimetry (DSC). The alloy strips with equiaxed and columnar grain structures were prepared via cold mold continuous casting and hot mold continuous casting, respectively. The process parameters for preparing billet via cold mold continuous casting are: melt temperature 1250 °C, mold temperature 20 °C, and casting speed 60 mm/min. The process parameters of hot mold continuous casting technology for preparing billet are: melt temperature 1250 °C, mold temperature 1150 °C, and casting speed 20 mm/min. Four high-rolling mills are used to cold roll the two types of strip billets with deformation of 25%, 50%, 75%, and 98%, respectively, and no heat treatment is carried out during rolling. The schematic diagram of strip preparation and cold rolling process is shown in Figure 1.

### 2.2. Microstructure Analysis

After rough grinding, fine grinding, and mechanical polishing, the sample was corroded by FeCl_3_ + HCl + H_2_O mixed solution, and the metallographic structure of the sample was observed under the OLYMPUS-BX53M optical microscope. The above samples were electropolished with an electrolyte consisting of H_3_PO_4_ + C_2_H_5_OH + CH_3_CH_2_CH_2_OH + H_2_O. The microstructure and crystal orientation of the samples were detected by EBSD using a Supra 55 scanning electron microscope (SEM). A longitudinal cross-section sample with a thickness of 0.5 mm was taken from the strip and grounded to 30–50 μm, the sample was double-jet electropolishing with a solution of HNO_3_ + CH_3_OH at −30 °C with a current of 45 mA. The microstructure of the samples was observed by Transmission electron microscopy (TEM) of JEM-2010. The phase structure of the samples was analyzed by Bruker D8 Advance X-ray diffrotometer with CuKα radiation at 40 kV and 40 mA. The dislocation density and microstrain of the samples were calculated by using Jade5.0 software according to the XRD patterns.

### 2.3. Mechanical Properties and Residual Stress Test

The tensile specimen was processed from the strip by wire cutting. According to GB/T 228-2010 standard (metallic materials-tensile testing-method of the test at room temperature, 2010), the tensile test was carried out on an MTS810 electronic universal testing machine with a strain rate of 1.0 × 10^−3^ s^−1^. According to GBT 7704-2017, the surface stress was measured by XRD residual stress technique. According to the principle of XRD stress measurement method Sin^2^*ψ*, the residual stress is measured with a PANACO Xpert MRD diffractometer, and the isocline method is used to measure residual stress the under the conditions of the copper target, 40 kv, 40 mA. 

## 3. Results and Discussion

### 3.1. Solidification Structure of C70250 Copper Alloy Strip and Surface Quality of Cold Rolled Strip

Solidification microstructures of C70250 copper alloy strip billets prepared by cold mold continuous casting and hot mold continuous casting technology are shown in Figure 2. The two strips are called equiaxed grain (EG) and columnar grain (CG) strip billets respectively. The crystal orientation of equiaxed grain strip billets is randomly distributed, and the large angle grain boundaries are predominant among the grains. The microstructures in the columnar grain strip are mainly grains growing along the traction direction. The grains own preferred <001> orientation along the axial direction, the difference of orientation between grains is small, and the grain boundaries are mainly at small angles. Cracks and peeling appear on the surface of the equiaxed grain strip when the deformation amount is 98%. When the deformation amount of the columnar grain strip is 98%, no crack appears and the surface quality is lighter. The morphology of the two large deformation cold rolled strips is shown in Figure 3.

### 3.2. Mechanical Properties and Residual Stress of Cold Rolled C70250 Copper Alloy Strip

As shown in Figure 4. The tensile strength of the EG strip increases from 288 MPa as cast to 646 MPa with 98% deformation, and the elongation decreases from 28.0% as cast to 1.1% with 98% deformation. The tensile strength of the CG strip increases from 274 MPa as cast to 601 MPa with 98% deformation, and the elongation decreases from 31.6% as cast to 3.2% with 98% deformation. The residual stress of the EG strip rises slowly from 62 MPa as cast to 75 MPa with 75% deformation and then rises sharply to 372 MPa when the deformation amount rises to 98%. The residual stress fluctuates with the increase of cold rolling deformation for the CG strip. When the deformation amount reaches 98%, the residual stress of the CG strip is only 125 MPa, which is about 1/3 of the EG strip. It was found that the tensile strength and residual stress of CG strips are lower when compared to the EG strip after cold rolling with the same large deformation amount.

### 3.3. Microstructure of Cold Rolled C70250 Copper Alloy Strip

To discover a mechanism of columnar grain structure that improves the cold working performance of strips compared with equiaxed grain structure, the changes in the microstructure of two strips during cold rolling were compared and analyzed. Figure 5 shows the microstructure of two strips of cold rolling, deformed by a 25% reduction. RD, ND, and CD show rolling, normal, and casting directions. There are many deformation lines and the distribution of deformation lines is not uniform in the EG alloy strips. The columnar grain of the CG strip becomes flat, and no obvious shape trace is produced inside the grain.

Figure 6 shows the internal dislocation structure observed by TEM when the deformation amount is 25%. It can be seen from the figure that a dense planar dislocation structure is mainly formed inside the strip. Many superposition phenomena of dislocation occur inside the EG strip, and large dislocation density leads to the entanglement of dislocation lines, as shown in Figure 6a. The planar dislocations inside the CG strip are significantly reduced, as shown in Figure 6b. The above results show that when the cold rolling deformation amount is 25%, the plastic deformation is mainly caused by the dislocation plane slip mechanism in the strip. Compared to the EG strip, the CG strip has less dislocation slip.

When the deformation amount of the strip is 50%, shear zones appear in both strips, which only develop inside the grain and fail to cross grain boundaries, as shown in Figure 7. The distribution of the shear zone in the EG strip is not uniform, the direction is not consistent, and local uneven deformation becomes more severe. The shear bands of the CG strip are evenly distributed in the same direction, and the angle is 30° to the rolling direction.

TEM analysis of the cold-rolled strip with a 50% reduction deformation showed that we can not only observe the dislocation structure similar to that shown in Figure 6 but also observe the obvious stacking fault structure. It can be seen from Figure 8 that there are two groups of stacking fault structures in the EG strip. The fault planes are $\left(02\bar{2}\right)$ and (311) crystal planes respectively, and they cross each other. The internal stacking faults of the CG strip are distributed in parallel along the same direction, and the stacking fault plane is the $\left(\bar{3}\bar{3}1\right)$ crystal plane. The above results show that when the deformation amount of the strip reaches 50%, the shear zone and stacking fault also occur in the process of cold rolling besides dislocation slip, and the deformation of the columnar grain strip is still relatively uniform.

When the cold rolling deformation of the strip increases to 75%, the number of shear bands in the two alloy strips increases significantly, as shown in Figure 9. The shear band of EG strips is denser, and the direction of shear bands inside each grain is still different. Some shear bands pass through the grain boundary and intersect with each other, and there are obvious cutting traces of grains, as shown in Figure 9a. Under the same deformation, there are relatively few shear bands in each grain of the CG strip, which are still mainly distributed in parallel along the same direction. The included angle between the shear band and the rolling direction is about 20°, and the grain boundary is still clear, the shear band can also cut through the grain boundary with a small angle, as shown in Figure 9b.

TEM analysis of the cold-rolled strip with a 75% reduction shows that a lot of deformation twins are generated in the EG strip, as shown in Figure 10. Deformation twin structure is also observed in the CG strip. The critical stress required to induce twin deformation is higher than the critical stress required for slip deformation, and dislocation slip will occur earlier than twin deformation [19]. The above results show when the slip deformation of the strip is hindered in the process of cold rolling, the twins will often change their hard orientation, which makes it difficult to continue to deform into a soft orientation, after which the deformed twins appear near the dislocation entanglement, which is more prone to subsequent slip deformations.

When the deformation amount increases to 98%, the intersecting shear bands inside the EG strip cut the Cu matrix structure into multiple diamond areas, and the internal grains are obviously broken, as shown in Figure 11a. When the deformation amount of the CG strip is 98%, the cold-rolled sample forms a fiber strip structure, and the shear band has thoroughly disappeared, as shown in Figure 11b. TEM analysis of the cold-rolled strip with 98% deformation amount shows that a large number of sub-grains are formed in the EG strip and the selected area electron diffraction pattern is annular, indicating that the microstructure is composed of sub-grains with random orientation, and the size of sub-grains is about 80 nm, as shown in Figure 12a. A fiber strip structure is formed inside the CG alloy strip, as shown in Figure 12b.

The two alloys were analyzed by XRD, and the microstrain and dislocation density were statistically analyzed by Jade5.0 software. The microstrain [20] was estimated by Formula (1).
(1)$$S=\frac{FW(S)}{4\tan(\theta )}$$
where *FW*(*S*) is the calculated maximum half-peak width (Rad) for the XRD peak, and *θ* is the corresponding diffraction angle.

On the basis of the optimized Williamson-Hall equation [21], the density of dislocation was estimated by Formula (2).
(2)$$\rho =A(2\cos\theta_{B}\Delta \theta /\lambda -0.9/d{)}^{2}$$

In the formula, *ρ* is the density of dislocation inside the metal, *A* is the fixed constant value of the metal, and 2Δθ is the half-width of the XRD diffraction peak, which increases along with the increase of dislocation density, d is the average grain size, λ is the wavelength of the XRD, and $\theta_{B}$ is the angle of Bragg reflection.

Table 1 summarizes the estimated results of the two strips under different deformation amounts. The initial lattice parameter of Cu without treatment is a = 0.3615 nm [22]. The average lattice parameters of the samples increase after cold deformation, indicating that there are high internal stresses in these samples. By comparing the micro strain and dislocation density in Table 1, it can be seen that the dislocation density and micro strain gradually rise, indicating that the level of lattice deformation and distortion in the alloy gradually increases. The dislocation density and micro strain of the CG strip are always less than that of the EG strip, which also proves that the CG strip has better cold working deformation ability than the EG strip.

Figure 13 shows the SEM fracture of the C70250 copper alloy CG strip undergoing varying degrees of deformation. The fracture of the stretched sample still shows the typical dimple fracture characteristics, which indicates that the continuous directional solidification alloy with columnar grain microstructure has good machinability. According to the above experimental results and analysis, the C70250 copper alloy strip produced by hot mold continuous casting has the good surface quality and can be directly rolled without surface treatment. The as-cast alloy has stronger axial columnar grains and higher elongation after fracture (31.6%), and the maximum cumulative cold deformation can reach 98%. On this basis, this paper proposes a short process of continuous directional solidification→direct rolling to produce a copper alloy thin strip, which eliminates hot rolling, surface treatment, intermediate annealing, pickling, and other processes.

### 3.4. Improving Mechanism of Cold Working Performance of C70250 Copper Alloy

In conclusion, plastic deformation mechanisms of C70250 copper alloys under different microstructures can be generalized in Figure 14. When the dissolution shear stress, dislocation, and slip are greater than the corresponding critical shear stress, the slip system is activated and plastic deformation occurs. The cold rolling deformation behavior can be segmented into three processes: First, initial microstructures are subjected to small compression deformation (25%), and the slip in the dislocation plane occurs, however, the columnar grain strip produces more uniform slip deformation than the equiaxed grain strip; Secondly, when the deformation degree reaches 50–75%, belonging to medium deformation, the dislocation density reaches the critical value, the critical shear stress of dislocation slip increases considerably so that dislocation slip is difficult to proceed. The stacking fault and shear band are produced through further severe plastic deformation, and the number of shear bands increases significantly with the increase of deformation. Deformation twins are formed after the stress reaches the critical stress of twin deformation. Due to the uneven deformation of equiaxed grain billets, the shear stress of grains is larger than that of columnar grain billets, which makes it easier to produce deformation twins; in the third stage, when the deformation degree is very large (98%), the equiaxed grain strip cuts the matrix due to the intersection of shear bands and stacking faults, and the grains are broken to form nanocrystals. Due to the uniform deformation structure, the columnar grain strip does not intersect shear bands and stacking faults, and finally forms a fibrous banded structure. The cold rolling deformation mechanism of the C70250 copper alloy strip with EG and CG structure is shown in Figure 14.

The elongation after fracture of the columnar grain strip is always greater than that of the equiaxed grain strip, which is mainly due to the fact that the columnar grain grows along the continuous casting direction, there are few transverse grain boundaries, and the grains have strong <001> orientation. Most grains are small-angle grain boundaries distributed along the continuous casting direction, which can reduce the number of grain boundary constraints and facilitate the coordinated deformation between grains. Dislocations and shear bands can continue to move directly through the grain boundary. Plane dislocation, stacking faults, and shear bands formed in the cold rolling process of columnar grain strip were evenly distributed, and the degree of stress concentration is small, which is conducive to the deformation of the material. During the rolling process of an equiaxed grain strip, it is easy to form intersecting plane dislocations and shear bands and other uneven structures, which makes it easy to produce stress concentration at the intersection of shear bands, resulting in cracks when the material is further deformed, and the plasticity is low.

The internal residual stress of alloy after cold rolling is mainly related to the uneven deformation produced in the rolling process. According to the scale of uneven deformation, residual stress can be divided into three categories: the first is the macro internal stress in the material, the second is the internal stress in several grains, and the third is the internal stress in several atoms. During the cold rolling process of equiaxed grain strip, it is easy to form intersecting plane dislocation and shear band structure. When the deformation amount reaches 50%, the shear deformation is hindered by the grain boundary and produces a large stress concentration at the grain boundary, the grains begin to distort. When the cold rolling deformation continues, the shear bands cross and the alloy will easily produce the second type of residual stress. If reduction continues to increase, the grain length of equiaxed grain still does not increase significantly, the grain width further decreases significantly, the degree of lattice distortion becomes more serious, and the residual stress continues to increase. When the deformation amount of equiaxed grain strip is 98%, the degree of lattice distortion and grain boundary density increases significantly due to grain breakage, which further leads to a significant increase in residual stress. When the cold rolling deformation amount of the columnar grain strip is small (0–50%), the grain boundary density basically remains unchanged, and the residual stress basically does not change. However, when the cold rolling deformation of the columnar grain strip is greater than 75%, due to the uniform deformation between columnar grain grains, although the cold working deformation also leads to the increase of lattice distortion of columnar grain strip, the columnar grain strip still mainly forms parallel plane dislocations, deformation twins and shear bands, the uniformity of intergranular deformation is good, and the second type of residual stress is small.

## 4. Conclusions

(1) C70250 copper alloy strips with columnar grain structure and equiaxed grain structure were prepared by hot mold continuous casting technology and cold mold continuous casting technology respectively. After cold rolling with 98% reduction, visible cracks and peeling phenomena appear on the surface of the equiaxed grain strip, while no cracks appear on the surface of the columnar grain strip and the surface quality is good. The elongation of the columnar grain strip is still as high as 3.2%, which is 2.9 times greater than that of equiaxed grain alloy. The residual stress of the equiaxed grain strip is 363 MPa, which is 2.7 times greater than that of the columnar grain strip. The strong axial oriented grains at 31.6% elongation after fracture make the columnar grain C70250 copper alloy strip have excellent cold working formability.

(2) Due to the random orientation of grains, the equiaxed grain strip is easy to form intersecting plane dislocations, stacking faults, and shear bands during cold rolling, and grain breakage occurs during cold rolling with large deformation; The columnar grain boundary orientation difference is small, and the grain orientation degree is high, the columnar grain strip forms parallel plane dislocation, stacking fault and shear bands in the cold rolling process, the deformation structure is uniform, and columnar grain strip finally forms the fibrous structure along the rolling direction, the elongation of columnar grain strip after large deformation cold rolling is greater than that of equiaxed grain strip.

(3) In the cold rolling process, the deformation of the columnar grain strip is more uniform than that of the equiaxed grain strip, under the condition of large plastic deformation, the grains of the equiaxed grain strip are broken, and the degree of lattice distortion is significantly increased, while the columnar grain strip produces uniform shear deformation, forming fibrous structure and greatly reducing residual stress.

## Figures and Tables

**Figure 1 materials-15-08724-f001:**
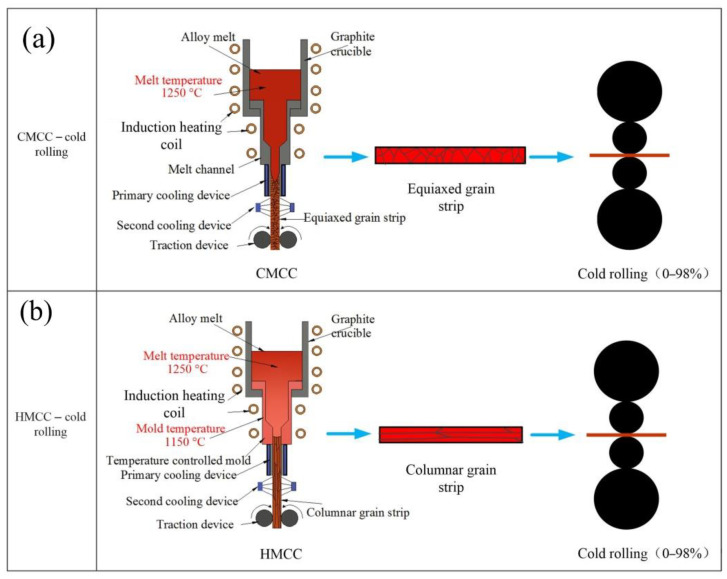
Schematic diagram of strip preparation and cold rolling process. (**a**) CMCC-cold rolling; (**b**) HMCC-cold rolling.

**Figure 2 materials-15-08724-f002:**
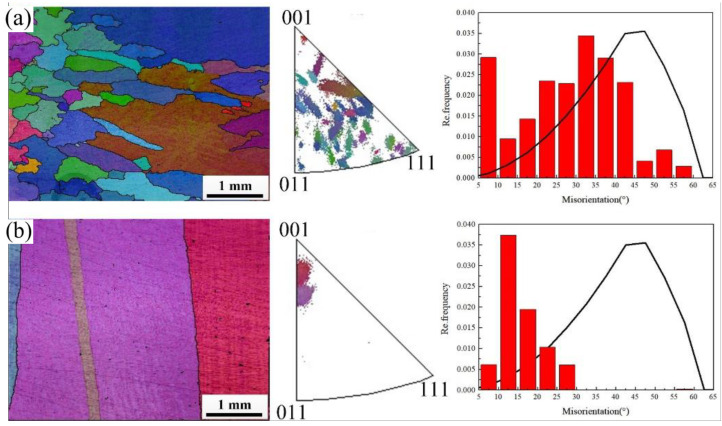
The casting microstructure morphology and crystal misorientation of C70250 copper strips. (**a**) EG strip and (**b**) CG strip.

**Figure 3 materials-15-08724-f003:**
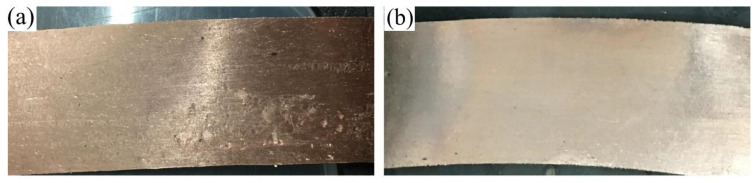
Surface morphology of C70250 copper alloy strip with different microstructure under 98% cold rolling deformation. (**a**) EG strip and (**b**) CG strip.

**Figure 4 materials-15-08724-f004:**
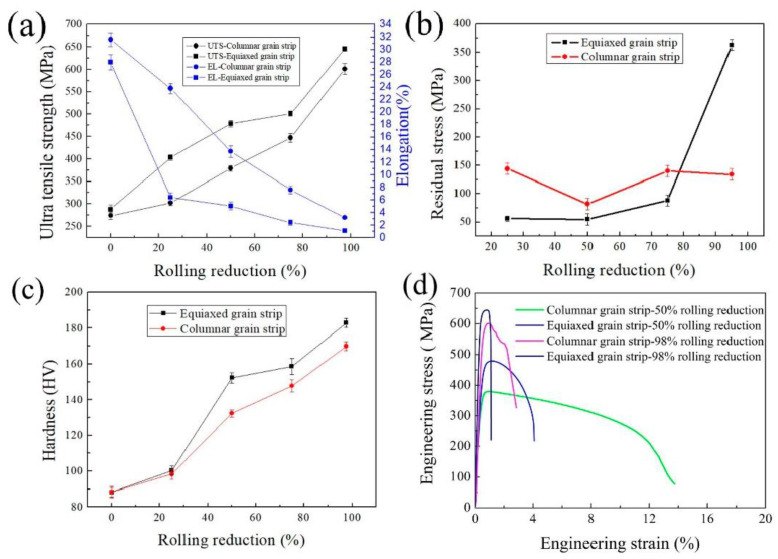
Mechanical properties and residual stress of C70250 copper alloy during cold rolling. (**a**) Strength and elongation; (**b**) Residual stress; (**c**) Hardness; (**d**) Engineering stress-strain curve.

**Figure 5 materials-15-08724-f005:**
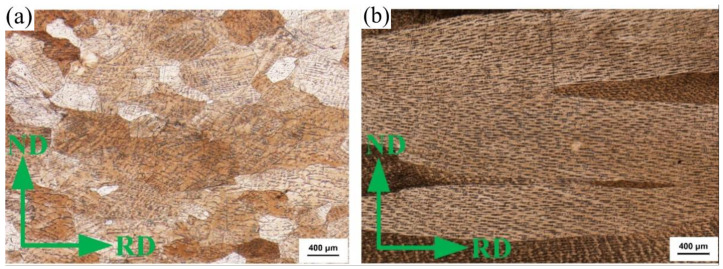
Metallography of the longitudinal section of cold rolled strip (ε = 25%). (**a**) EG strip and (**b**) CG strip.

**Figure 6 materials-15-08724-f006:**
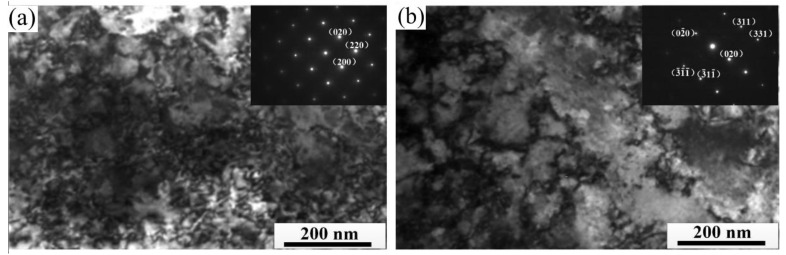
Dislocation structure of cold rolled strip (ε = 25%) (**a**) EG strip; (**b**) CG strip.

**Figure 7 materials-15-08724-f007:**
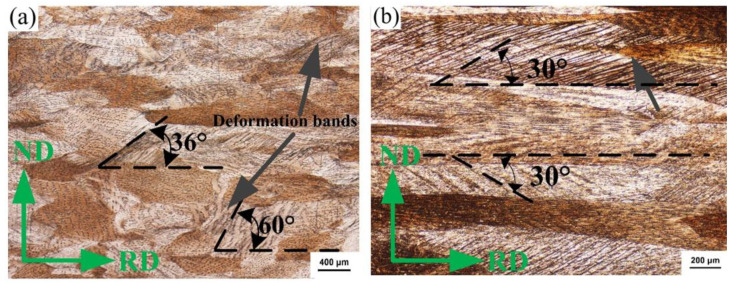
Metallography of longitudinal section of cold rolled strip (ε = 50%). (**a**) EG strip and (**b**) CG strip.

**Figure 8 materials-15-08724-f008:**
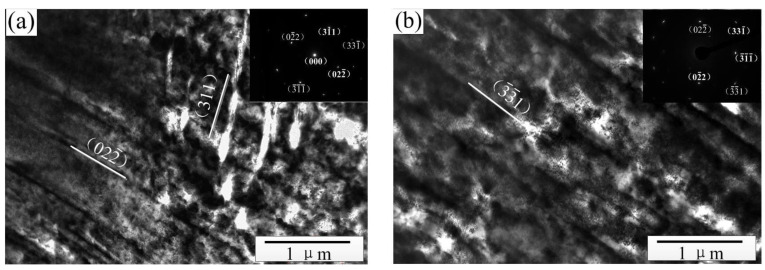
Fault structure of cold rolled strip (ε = 50%). (**a**) EG strip and (**b**) CG strip.

**Figure 9 materials-15-08724-f009:**
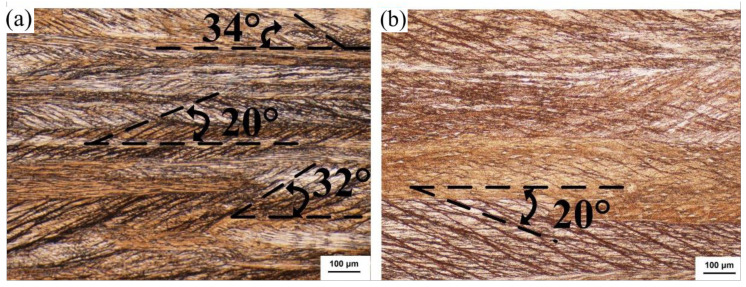
The longitudinal section metallography of cold-rolled strip (ε = 75%). (**a**) EG strip and (**b**) CG strip.

**Figure 10 materials-15-08724-f010:**
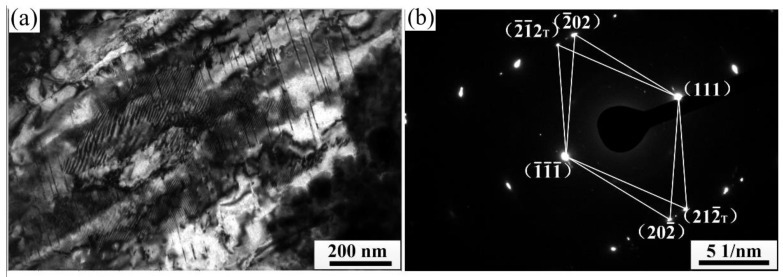
Deformation twin structure in cold rolled EG strip (ε = 75%). (**a**) Deformation twin and (**b**) Corresponding selected electron diffraction pattern.

**Figure 11 materials-15-08724-f011:**
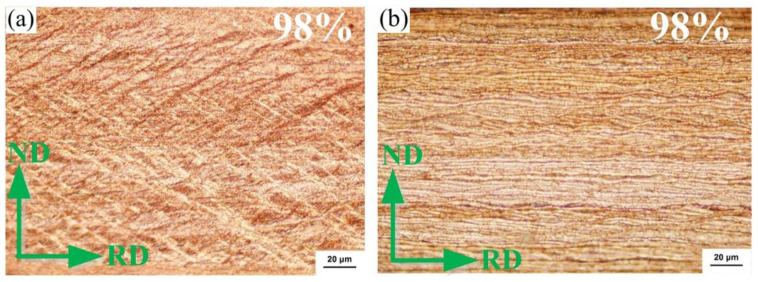
Longitudinal section Metallography of large deformation cold rolled alloy strip (ε = 98%). (**a**) EG strip and (**b**) CG strip.

**Figure 12 materials-15-08724-f012:**
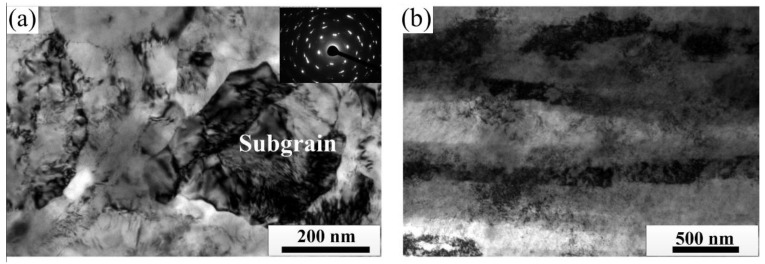
Grain structure of cold rolled strip (ε = 98%), (**a**) EG strip and (**b**) CG strip.

**Figure 13 materials-15-08724-f013:**
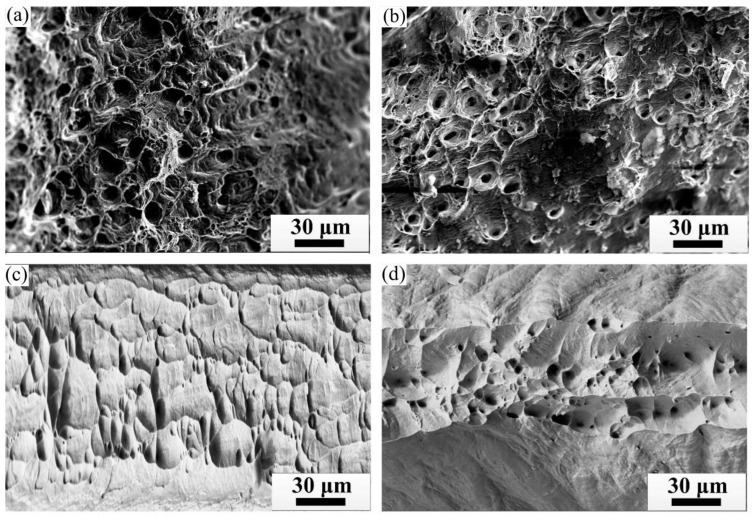
SEM fracture of CG strip undergoing different deformation. (**a**) 25%; (**b**) 50%; (**c**) 75%; (**d**) 98%.

**Figure 14 materials-15-08724-f014:**
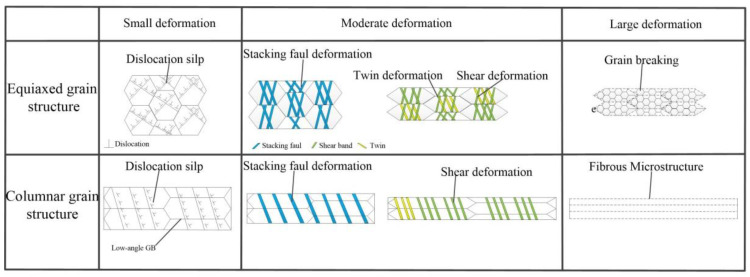
Deformation behavior of EG and CG strip during cold rolling.

**Table 1 materials-15-08724-t001:** Lattice parameters a, dislocation density *ρ*, and microstrain *S* of cold rolled alloys.

	Cold Deformation	*a* (nm)	*ρ* (×10 ^15^ m^−2^)	*S (*%)
Equaxied grain strip	0.25	0.36542	2.08	0.21
	0.5	0.37609	2.75	0.31
	0.75	0.38355	3.41	0.36
	0.98	0.38962	5.98	0.58
Columnar grain strip	0.25	0.36312	1.94	0.12
	0.5	0.37423	2.35	0.24
	0.75	0.37568	2.60	0.33
	0.98	0.38412	4.56	0.52

## Data Availability

The raw data required to reproduce these findings cannot be shared at this time as the data also form part of an ongoing study.

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
