# Peer review of "Effect and Mechanism of Solidified Microstructure on Deformation Behavior, Mechanical Properties, and Residual Stress of Cu-Ni-Si Alloy"

_materials, 2022, doi:10.3390/ma15248724_

Round 1
Reviewer 1 Report
The paper is well written. For all the test the authors must include the ASTM test conditions details in the manuscript. The authors choosen the composition ( Fixed ) on what basics. The general application of the selected compositions must be highlighted in the article.
In figure 2 the CG strip have cracks. Why? If possible the authors can include the base OM micro-structures with EBSD data. What is the nature to discuss about solidified structures as a seperate result and discussions
Author Response
Dear editor and the reviewers,
We thank the reviewer for his/her serious and pertinent comments. We have carefully read the comments and thought them over. The responses are listed as follows. Besides, the corresponded revisions are highlighted in red color in this revised manuscript.
Reviewer #1:
(1) The paper is well written. For all the test the authors must include the ASTM test conditions details in the manuscript. The authors choosen the composition ( Fixed ) on what basics. The general application of the selected compositions must be highlighted in the article.
Response:
Thank you for your suggestions. I have updated the test conditions in detail in the manuscript. The tensile specimen was processed from the strip by wire cutting. According to GB/T 228-2010 standard (metallic materials-tensile testing-method of test at room temperature, 2010), the tensile test was carried out on MTS810 electronic universal testing machine with a strain rate of 1.0×10−3s−1. According to GB/T 7704-2017, the surface stress was measured by XRD residual stress technique. According to the principle of XRD stress measurement method Sin2ψ, the residual stress is measured by panaco xpert mrd diffractometer, isocline method is used to measure residual stress under the conditions of copper target, 40kv, 40mA.
C70250 alloy is an age-strengthened Cu-Ni-Si alloy with high strength, high electrical conductivity and stress relaxation resistance. It is very suitable for lead frame alloy and high-density integrated circuit packaging. The general application of the selected compositions have been highlighted in red color in this revised
manuscript.
(2) In figure 2 the CG strip have cracks. Why? If possible the authors can include the base OM micro-structures with EBSD data. What is the nature to discuss about solidified structures as a seperate result and discussions.
Response:
Thank you for your suggestions. The crack of EG billet is mainly due to the large residual stress and poor workability in the process of cold rolling with large deformation. The crystal orientation in equiaxed grain strip billet is randomly distributed, and the large angle grain boundaries are predominant among the grains.
when the deformation degree is very large (98%), the equiaxed grain strip cuts the matrix due to the intersection of shear bands and stacking faults, and the grains are broken to form nanocrystals, which is easy to produce stress concentration at the intersection of shear bands, resulting in cracks when the material is further deformed, and the plasticity is low.
Sincerely,
We hope that our responses could meet the requirements of the reviewers and you. And if there are other questions about our revised manuscript, please don’t hesitate to contact us.
Best regards,

Reviewer 2 Report
The article “Effect and mechanism of solidified microstructure on deformation behavior, mechanical properties and residual stress of Cu-Ni-Si alloy” is devoted to the study of mechanical properties and microstructure of copper-based alloy and explanation of the mechanism for providing higher mechanical properties of microstructure with columnar grains. It is shown that due to the formation of columnar grains in the process of hot mold continuous casting, the kinetics of the dislocation structure formation changes with an increase of deformation (reduction). A change in the kinetics of the linear defects formation, as well as a change in the mechanism of their interaction with grain boundaries, leads to the formation of lower residual stresses in the columnar grains structure. The results presented in the paper have been proven using several research methods, including optical microscopy, scanning and transmission electron microscopy, diffractometry, and other methods.
The structure and filling of the article is correct, but the following remarks should be made.
1. Figure 4a shows the dependences of the mechanical characteristics of the material under study on deformation level during rolling. In addition to data on the ultimate tensile strength, it would be interesting to look at the values of the yield strength, as well as the nature of the material hardening during tension. It is possible to add two tension curves for material with equiaxed grains and columnar grains.
The scatter of elongation is small and does not exceed 1% (in absolute values) for the same samples. How many samples were tested that you managed to get such a small scattering?
Also in this figure, the scale of the elongation axis should be changed to avoid negative values on the axis.
2. Fractures at the deformation of 50% and 75% differ in structure (Figures 13b and 13c). What is the reason for this difference? Since a fractographic analysis was carried out, the approximate values of cross-section reduction after rupture may be presented for the samples with various reduction level. This parameter is also a good indicator of the material plasticity.
Minor remarks:
- Line 84. DCS - does it mean “Differential Scanning Calorimetry”? Please decipher in the text.
- Lines 84-85. At this point, the phrase that “strips with equiaxed and columnar grains were obtained by cold mold continuous casting and hot mold continuous casting” is repeated three times in the text, including the abstract, the end of the introduction, and section 2.1.
- The melting temperature in the text on lines 85 and 87 is 1250C, and in Figure 1 it is 1200C.
- The abbreviations RD, ND and CD should be explained in the text (Figures 5, 7, 11). What is the difference between CD and RD directions?
- A reference to Figure 8 should be added in the text.
_____
The article is a complete study, well formed, interesting, and easy to read. The results are useful from a practical point of view. It is also necessary to keep in mind that at any level of deformation (reduction), the structure with columnar grains remains anisotropic, so in the future it’s interesting to assess the anisotropy of the mechanical properties of such a material.
I believe that the article is worthy of publication in the "Materials" after the elimination of these minor remarks.
Author Response
(1) Figure 4a shows the dependences of the mechanical characteristics of the material under study on deformation level during rolling. In addition to data on the ultimate tensile strength, it would be interesting to look at the values of the yield strength, as well as the nature of the material hardening during tension. It is possible to add two tension curves for material with equiaxed grains and columnar grains.
The scatter of elongation is small and does not exceed 1% (in absolute values) for the same samples. How many samples were tested that you managed to get such a small scattering?
Also in this figure, the scale of the elongation axis should be changed to avoid negative values on the axis.
Response:
Thank you for your suggestions. The tension curves and hardness for material with equiaxed grains and columnar grains have been added. Three stress-strain curves for each sample were measured, and the overall fluctuation range was small. Besides, the scale of the elongation axis have be changed in Fig. 4(a).
(2) Fractures at the deformation of 50% and 75% differ in structure (Figures 13b and 13c). What is the reason for this difference? Since a fractographic analysis was carried out, the approximate values of cross-section reduction after rupture may be presented for the samples with various reduction level. This parameter is also a good indicator of the material plasticity.
Minor remarks:
- Line 84. DCS - does it mean “Differential Scanning Calorimetry”? Please decipher in the text.
- Lines 84-85. At this point, the phrase that “strips with equiaxed and columnar grains were obtained by cold mold continuous casting and hot mold continuous casting” is repeated three times in the text, including the abstract, the end of the introduction, and section 2.1.
- The melting temperature in the text on lines 85 and 87 is 1250C, and in Figure 1 it is 1200C.
- The abbreviations RD, ND and CD should be explained in the text (Figures 5, 7, 11).
What is the difference between CD and RD directions?
- A reference to Figure 8 should be added in the text.
Response:
Thank you for your suggestions. Fig. 13 shows the SEM fracture of C70250 copper alloy CG strip undergoing varying degrees of deformation. The fracture of the stretched sample still shows the typical dimple fracture characteristics, which indicates that the continuous directional solidification alloy with columnar grain
microstructure has good machinability. With the increase of the cold rolling rate, it can be seen that the dimple size and depth of the alloy decrease gradually, the dimple number decreases gradually, and the elongation of the alloy decreases gradually.
DSC means “Differential Scanning Calorimetry”, the corresponding
modification has been highlighted in red.
The melting temperature in the Fig. 1 is 1250 °C, and it has been modified.
The abbreviations RD, ND and CD have be explained in the text (Figures 5, 7,11).
The reference to Figure 8 have been added in the text.
Sincerely,
We hope that our responses could meet the requirements of the reviewers and you. And if there are other questions about our revised manuscript, please don’t hesitate to contact us.
Best regards,
